# Effect of Zinc Source and Level on Growth Performance and Zinc Status of Weaned Piglets

**DOI:** 10.3390/ani11072030

**Published:** 2021-07-07

**Authors:** Anna Szuba-Trznadel, Anna Rząsa, Tomasz Hikawczuk, Bogusław Fuchs

**Affiliations:** 1Department of Animal Nutrition and Feed Management, Wrocław University of Environmental and Life Sciences, J. Chełmońskiego 38D, 51-630 Wrocław, Poland; tomasz.hikawczuk@gmail.com (T.H.); boguslaw.fuchs@upwr.edu.pl (B.F.); 2Department of Immunology, Pathophysiology and Veterinary Preventive Medicine, Wrocław University of Environmental and Life Sciences, C.K. Norwida 31, 50-375 Wrocław, Poland; anna.rzasa@upwr.edu.pl

**Keywords:** zinc, zinc oxide nanoparticles, growth performance, diarrhea incidence, piglets

## Abstract

**Simple Summary:**

Diarrhea is common in weaned piglets. Feed supplementation with Zn is one strategy for preventing such disturbances. A total ban on the use of therapeutic doses of Zn will be introduced from June, 2022 in the European Union; hence, the evaluation of different lower doses and forms of this element is very important and challenging. This paper shows the effects of using different forms of Zn preparations (i.e., zinc sulfate, zinc oxide, and zinc oxide nanoparticles) on weaner rearing results and Zn balance. The obtained results indicate that low levels of zinc oxide nanoparticles can express a similar antidiarrheal action as high therapeutic doses of zinc oxide. In this way, the Zn dose concentration in feed supplements for pigs can be decreased. The application of nanoparticles can lead to measurable benefits, including a higher growth rate and better feed use.

**Abstract:**

The aim of this study was to evaluate the effect of zinc (Zn) supplementation in different commercial forms on the growth performance, health status, and Zn balance of weaners in field conditions. The animals were fed pre-starter (from the 28th to 47th day of life) and starter (from the 48th to 74th day of life) mixtures differing in Zn form and concentration. Group I was given ZnSO_4_ at 150 mg kg^−1^; Group II received pre-starter zinc oxide (ZnO) at 3000 mg kg^−1^ and starter at 150 mg kg^−1^; and Group III was given 150 mg kg^−1^ of zinc oxide nanoparticles (nZnO). We found that the average daily gain in Group I was significantly lower, compared to Groups II and III. A commonly accepted level of Zn (150 mg kg^−1^) as nZnO can be recommended, instead of therapeutic doses of Zn preparations with the same efficiency. Moreover, a lower level of Zn in the diet can prevent the excessive accumulation of this element in waste and, thus, reduce environmental damage.

## 1. Introduction

The total ban on the use of antibiotics as growth promoters in animals, introduced in the European Union from 1 January 2006 (Regulation (EC) No, 1831/2003) [1], and tremendous pressure to reduce their therapeutic usage, has resulted in extensive research on the use of various feed additives [2]. These can modulate or protect the immune system and influence the gut microbiota composition of animals, hence impacting the host’s health status, production efficiency, and performance [3]. This group of additives consists of a large number of products, including probiotics, prebiotics, symbiotics, acidifiers, yeast products, preparations isolated from plants, and different forms of zinc (Zn) [3,4,5,6,7].

As a trace element, zinc supports important functions in animal organisms, influencing growth, immunity, development, and reproduction [8]. Zinc compounds positively affect the digestive tract through increasing mucosal thickness, villi height, activity of enzymes, and regulation of digestive tract micro-organisms [9,10,11,12].

In practice, to reduce intensive diarrhea in piglets, pharmacological doses of ZnO (1000 to 4000 mg kg^−1^) are prescribed. However, the allowed maximum of 150 mg kg^−1^ is still higher than the requirements of weaned piglets [13]. The requirement for Zn supplementation in the diet of weaned piglets decreases from 100 to 60 mg kg^−1^ when their body weight increases from 5 to 50 kg. However, due to the effect of growth factors or large safety margins, more than 10 to 20 times the required inorganic Zn—such as zinc oxide (ZnO)—is commonly applied in their diet, in order to maximize the performance of piglets after weaning [14].

Additionally, the excess excreted Zn adversely affects the environment. An excess of Zn can also be accumulated in tissue (food of animal origin), which has been connected with increased cellular stress and lower absorption of other micro- and macro-elements [9,15,16,17]. Therefore, the European Union’s directive decreased the maximum contents of micro-elements to be used as feed additives (Commission Regulation EU No. 1095/2016 of 6 July 2016) [18]. At present, in the European Union, zinc oxide can only be applied at 150 mg kg^−1^ dosages as a nutritional component for piglets; however, from June of 2022, therapeutic doses of ZnO will be banned in animal feeds [19].

Many researchers have focused their attention on new sources and forms of Zn, in order to protect the environment, animal health, and quality of the final products, with the intent of improving Zn use, availability, and absorption from the intestinal content, thus reducing the amount of this additive in animal diets.

The above intentions can be met by nanoparticles created through nanotechnological processes. Due to nanometric dimensions being less than 100 nanometers (nm), ZnO nanoparticles (nZnO) are characterized by special physical and chemical properties, including the ability to cross most biological barriers (to date, these mechanisms have not been elucidated). The use of nZnO as a feed additive increases the efficacy and assimilability of zinc in the digestive tract, which allows for limiting its content in feed [20]. Nano-ZnO has exhibited a higher antibacterial activity than ZnO, as well as demonstrating a mutagenic effect on bacteria (e.g., *E. coli*), by causing cell membrane deformation [21,22,23].

The idea of investigating this issue arose following diarrhea incidents at one farm and a veterinarian’s decision to resolve the problem by introducing nutritional treatment with different forms of Zn additives. We evaluated the effects of supplementing weaners with different commercial forms of zinc on growth performance, health status, and zinc balance in field conditions. Moreover, we check if the use of nZnO at an acceptable level (150 mg kg^−1^) can effectively counteract diarrhea incidents.

## 2. Materials and Methods

### 2.1. Experimental Design and Animal Housing

The experiment was performed in field conditions at a farm working in a closed cycle located in the western part of Poland, a region with a high density of pig production. Post-weaning diarrhea (PWD) was a persistent problem there.

The study was carried out on 300 crossbred (Polish Large White × Polish Landrace) weaners, divided into three experimental groups according to initial body weight (8.3 ± 0.4 kg). They were subject to routine care and veterinary treatment performed on the farm. After weaning (28th day of life), the animals were moved to a nursery unit and kept in slatted pens in groups of 10 pigs. The animals were reared in a litter-free system with optimal humidity (60%) and temperature (+19 °C) levels, which were automatically controlled, in one building. The experiment ended on the 74th day of life of the animals, when they were moved to the fattening sector.

### 2.2. Source of Tested Product and Experimental Feed Mixtures

The weaners were fed a pre-starter diet from the 28th to 47th day of life and a starter diet from the 48th to 74th day of life. They were offered feed mixtures ad libitum from self-feeders, and had free access to water by nipple drinkers with controlled flow, adjusted to their needs.

The experimental groups were differentiated in terms of concentration and form of zinc supplemented in the feed. Group I received a standard pre-starter, followed by a starter diet with the addition of zinc sulfate (35%; at the level of 150 mg kg^−1^ in each feed type). Group II was given a pre-starter feed mixture containing zinc oxide (75%) at 3000 mg kg^−1^ (as a medicated feed, which was permitted under the supervision of a veterinarian), then a starter feed with ZnO at 150 mg kg^−1^. Group III was fed mixtures supplemented with nZnO (99%; at the level of 150 mg kg^−1^ in the pre-starter and starter feed mixtures). We used a Nano-Zn preparation from Dashinou Nanotechnology (Changzhou, China). According to the producer, the purity of this product was 99%, the mean size of nanoparticles varied between 10 and 30 nm, and it was obtained using a mechanical method.

All feed mixtures were produced in the form of a fine granulate by a professional feed manufacturer (the feed manufacturer’s identification number was αPL3029001p). The pre-starter and starter feed mixtures were characterized by parameters complying with the experimental design, and were prepared in accordance with nutritional requirements defined by the Polish Nutrient Requirements for Swine [24]. The contents of basic nutrients in diets were determined according to the AOAC International Official Methods of Analysis [25], including dry matter (DM; AOAC: 934.01), crude protein (CP; Kjeldahl method, AOAC: 984.13), crude ash (CA; AOAC: 942.05), and ether extract (EE; Soxhlet method, AOAC: 920.39A) with the use of a BUCHI Extraction System B-811 (BÜCHI, Flawil, Switzerland), and crude fiber (CF; Henneberg and Stohmann method, AOAC: 978.10). The composition and nutritional values of the diets are presented in Table 1 and Table 2.

### 2.3. Sampling and Measurements

During the experiment, animals were weighed (BW) on the 28th, 47th, and 74th day of life. Feed intake per animal was measured every day, as the total amount of feed delivered to each pen in two nutritional periods, minus the amount of feed remaining in the feeders divided by the number of pigs in the pen. To calculate FCR, the total feed intake was divided into weight gain in the two nutritional periods. Mortality was considered in the calculations of DFI and FCR, as the FI and BW of dead animals were registered. Losses of animals, diarrhea incidents, and feces consistency and color were also recorded. The feces consistency was evaluated within each pen every day in the morning. Evaluation of fecal consistency was determined based on a 5-degree scale: hard (1), compact (2), normal (3), soft (4), and watery (5). The occurrence of diarrhea was defined when the feces were thin and watery. The diarrhea rate (%) for the pigs was calculated according to the following formula [26]:(1)$$\begin{array}{l} \mathrm{Diarrhea}\;\mathrm{rate}\\ =\frac{number\;of\;pigs\;with\;diarrhea\times number\;of\;days\;of\;diarrhea}{total\;number\;of\;pigs\times total\;number\;of\;days\;of\;experiment}\times 100 \end{array}$$

During the five-day collection period (from the 42nd to 47th day of life), samples of feces were collected every day (four times) from each pen, and then sealed in plastic bags and kept frozen (at –20 °C) until further analysis. At the end of the collection period, feces samples were thawed, pooled, and an average sample from each pen was separately prepared from the whole collected material.

Additionally, during routine health control performed at the farm by the veterinarian on the 47th day of life, the blood was drawn from the *vena jugularis externa* using 5 mL polypropylene disposable syringes (PolFast, Siedlce, Poland) with 16–19 G × 1 ½” needles (MIFAM). Next, a portion of the blood sample, collected to obtain serum, was transferred to tubes with pearls, left for two hours and then centrifuged (MPW-223e laboratory centrifuge, Warszawa, Poland) at 2000× *g* for 10 min at room temperature. Next, for experimental needs, a part of the sampled serum (from seven randomly chosen piglets/experimental group) was taken, in order to determine the chosen parameters.

### 2.4. Laboratory Analysis

The following were determined in sera: total protein (TP) using the burette method, and its fractions (i.e., albumins and α-, β-, and γ-globulins) using the filter paper electrophoresis method; haptoglobin (Hp) using the guaiacol (peroxidase) test [27]; and alkaline phosphatase (ALP) activity, glucose (PAP), and urea by a colorimetric method with the use of a Pentra 400 analyzer (Horiba ABX Diagnostics, Montpellier, France) and Pentra reagents (ALP, Cat. No. A11A01654; PAP, Cat. No. A11A01668; Urea CP, A11A01641).

Zn contents were determined using Mars 5 Xpress Technology and a Fast Sequential Atomic Absorption Spectrometer Varian AA240FS (Santa Clara, CA, USA). For analysis, three representative samples of each feed (0.1–0.2 g each), as well as serum (1 mL), were mineralized with a Mars 5 version 194A06 (CEM Corporation, Matthews, NC, USA) microwave mineralization system. Samples were introduced into a burning crucible and 7 mL of ultra-pure nitric acid (HNO_3_) was added. Mineralization was performed according to a temperature program consisting of two stages: first stage—power 1600 W, temperature 140 °C, time 8 min; and second stage—power 1600 W, temperature 200 °C, time 10 min. Then, samples were poured into a 50 mL flask and topped up with deionized water. Zn contents in serum after gradual drying were also mineralized. The frozen feces samples (at a temperature of −20 °C for 24 h) were lyophilized in an Edwards Modulyo (Bristol, UK) analytical lyophilizer (under pressure, at a temperature of −55 °C, for the required time to obtain a constant mass sample). After lyophilization, the samples were ground to powder, for 2 min, in a WŻ-type laboratory cutting mill produced by Zakład Badawczy Przemysłu Piekarskiego Sp. z o.o. (Bydgoszcz, Poland). The analytical determination of Zn in samples was performed by using flame atomic absorption spectrometry (FAAS). An air-acetylene oxidizing flame was applied for the atomization of Zn. All measurements were carried out in triplicate using the following instrumental conditions: wavelength, 213.9 nm; slit width, 1 nm; air flow, 13.5 L·min^−1^; acetylene, flow 2.00 L·min^−1^; PROMT mode; background correction, yes.

The amount of Zn determined in feces served as the basis for calculating the zinc balance. Based on the obtained results, the amounts of zinc intake in the diet, excreted in the feces, and retained in the body were calculated for each replicate and presented in a table as a mean value with standard deviation. The apparent total retention (%) was calculated according to the following equation:(2)$$\mathrm{Apparent}\;\mathrm{total}\;\mathrm{retention}=\frac{Zn\;intake-Zn\;excretion}{Zn\;intake}\times 100$$

### 2.5. Statistical Analysis

All data are expressed as means ± SD (standard deviation). The results, calculated as the means for each pen, were analyzed by one-way ANOVA using the Statistica 12 statistical package [28]. Differences between means were determined by Duncan’s post hoc test, and *p* ≤ 0.05 and *p* ≤ 0.01 were taken to indicate various levels of statistical significance.

## 3. Results

### 3.1. Rearing Performance of Weaned Piglets

Table 3 presents the production results in the two feeding phases. In the post-weaning period, a tendency towards variation of average BW among groups was observed. Analysis of the whole experimental period revealed an improvement in ADG, which were 5% and 10% higher in Groups II and III compared to Group I. These differences (in BW and ADG) were confirmed statistically (*p* < 0.05).

There were no significant differences between Groups, in terms of FI, but significant differences were found in FCR. The lowest FCR was recorded in Group III, and this result significantly differed from those obtained in Groups II and I in the first feeding phase (pre-starter mixtures; *p* < 0.01) and in the second feeding phase (starter mixtures; *p* < 0.05). Considering the whole rearing period (from 28th to 74th day of life), the lowest FCR was noted in Group I.

Diarrhea incidents occurred in Group I three times more often than in Groups II and III. In Group I, the feces were black, tarry, and distinctly loose. In contrast, feces from Groups II and III showed a typical normal grey–brown color with consistency scored as 3 points (i.e., typical of healthy animals).

Over the whole rearing period, the highest losses were noted in Group I—more than three times higher, when compared to Groups II and III. Higher losses took place in the second feeding phase in all groups. The animals in Group I were exhausted by diarrhea and distinctly differed from other pigs, in terms of body condition.

### 3.2. Biochemical Indices and Zinc Status

The analysis of biochemical indices (Table 4) determined in blood serum found no significant differences, in terms of concentration of TP, albumins, globulins, glucose, and Hp, among the experimental groups. The activity of ALP was higher in Groups I and III, in comparison to Group I, by 40% and 49%, respectively.

In Groups I and II, significantly higher concentrations of urea in blood were recorded, in comparison to that in Group III (*p* < 0.01). The highest concentration of Zn was noted in Group II; this result differed significantly from those obtained in Groups I and III (*p* < 0.01).

The highest Zn excretion in feces was noted in Group II, which differed significantly from those obtained in Groups I and III (*p* ≤ 0.01). The highest Zn retention was observed in Group II and the lowest in Group I, where the difference (23.9%) between those Groups was significant (*p* ≤ 0.05).

Zn intake of animals in Group II was 14 times higher, on average, while excretion ranged from 9 to 11 times higher; thus, Zn retention was 30 and 18 times higher, compared to Groups I and II, respectively (*p* < 0.05). Comparison of the retention of this element with FI unequivocally demonstrates that daily retention (expressed in %) was about 1.5–2 times higher in the group receiving therapeutic doses of ZnO. The highest excretion was noted in Group I, while the lowest was in Group II. Excretion of zinc (expressed in %) was lower in experimental Groups II and III. Conversely, in the case of animals offered the highest dose of Zn, more than 50% of this element was excreted into the environment.

## 4. Discussion

Dietary supplementation with different levels of ZnO (3000 m kg^−1^) and nZnO (150 mg kg^−1^) improved the growth performance of piglets. It is commonly known that, in pigs, feeding pharmacological levels of zinc oxide can improve BW [29], ADG, and/or FI [30,31]. Estienne et al. [32] noted that such levels enhanced nursery growth performance during only the early post-weaning period. The results obtained by Wang et al. [33] showed that 800 mg kg^−1^ nZnO might serve as a potential substitute for 3000 mg kg^−1^ ZnO in the diet of weaned piglets. The same tendency was confirmed in our study. The most notable observation is that supplementation with nZnO (150 mg kg^−1^) could improve ADG and FCR as efficiently as using 3000 mg kg^−1^ of conventional ZnO. Similar results have been reported by Pei et al. [34]. The obtained results are promising for pig producers, especially as a ban on therapeutic doses in pig feeding will be implemented. Taking into account that there are also suggestions to reduce the ZnO level to 100 mg kg^−1^, the effectiveness of such a dose should be still tested. A lower dose of ZnO in feed mixtures for weaned piglets could be considered. Li et al. [35] reported that nZnO at 120 mg kg^−1^ did not improve the growth rate in weaned pigs.

Analysis of diarrhea incidences showed that adding 150 m kg^−1^ of nZnO was as efficacious as 3000 mg kg^−1^ of ZnO. The similar conclusions were reported by Pei et al. [34] and Long et al. [36]. Furthermore, Sun et al. [37] noted that dietary supplementation with 400–600 mg kg^−1^ Zn (as nZnO) reduced the incidence of diarrhea.

The results of biochemical analyses of blood presented in Table 4 were in the physiological range [38]. The slightly higher concentration of TP and albumin in Group III could indicate a better nutrient supply in these animals and may reflect their more stable health status, which was confirmed, by lower concentration of Hp and standard deviation, in the frame of this Group [39]. Moreover, a significantly lower urea level indicates a better nitrogen balance in these animals. The study by Wang et al. [31] revealed a better dietary protein use by animals supplemented with ZnO and, consequently, a lower blood urea concentration vs. the control group (without dietary ZnO supplementation).

The activity of ALP remained in the upper range of normal values in Groups II and III, and were slightly higher than in Group I. Correspondingly, Mei et al. [40] and Revy et al. [41] demonstrated a tendency towards an increase in ALP level with increasing Zn concentration in the diet of animals. Furthermore, Sun et al. [37] and Wang et al. [42] reported that pharmacological ZnO supplementation improved ALP activity in sera of weaning piglets.

The highest serum Zn level in Group II was in agreement with the findings of other authors [31,43], who indicated that serum Zn concentration rose with ZnO content in the diet of weaners. Increase of dietary Zn doses above the requirements of animals also results in the increase of blood glucose level [44,45], which was confirmed in our study.

Although the growth rate of piglets can be improved by feeding pharmacological concentrations of ZnO, there are environmental concerns associated with it. The obtained results showed the retention of Zn in serum was significantly enhanced by the addition of a pharmacological dose of ZnO. Pei et al. [34] suggested that more Zn can be absorbed through the gastrointestinal tract if there is a high level of zinc in the diet. In the present study, excretion of Zn with feces was also associated with a high concentration of this element in the diet, confirming the results of other authors [46,47,48]. The results obtained by Pei et al. [34] indicated that nZnO (450 mg kg^−1^) may improve Zn absorption and reduce fecal Zn excretion, compared with a high level of ZnO (3000 mg kg^−1^).

## 5. Conclusions

A commonly accepted level of nZnO (150 mg kg^−1^) can be recommended, instead of therapeutic doses of Zn preparations with the same efficiency. Moreover, a lower dietary level of zinc can prevent the excessive accumulation of this element in waste and, thus, reduce environmental damage.

## Figures and Tables

**Table 1 animals-11-02030-t001:** Composition * and nutritive value of pre-starter mixtures for piglets.

Item	Group I (ZnSO_4_)	Group II (ZnO)	Group III (nZnO)
**Ingredients, %**			
ZnSO_4_	0.04	-	-
ZnO	-	0.04	-
nZnO	-	-	0.03
**Nutritional Value (% in 1 kg of Feed Mixture** **)**			
Metabolizable energy, 13.1 MJ; Crude protein, 16.5; Calcium, 0.64; P_total_, 0.54; Lysine, 1.15; Methionine, 0.43; Threonine, 0.75; Tryptophan, 0.24			
**Addition of Zn (mg in 1 kg of Feed Mixture)**			
Zinc (according to assumptions)	150	3000	150
Zinc (according to chemical analyses)	160	2700	160

* Basic composition of all mixtures: Barley, Wheat, Rice wafers, Soya bean meal (46% CP), Nuklospray L70 (mixture of dairy products), H-35 Fish pulp (DM 88%), Soya Hamlet Protein-300, Phoscal, Soy oil, Jelucel PF 200 (cellulose: content crude fiber > 98%), L-lysine (98%), Premix (0.5%) **, Bilo-Cid Bufor [acidifying mixture, composition: phosphoric acid (41%), formic acid (24%), cytric acid (11%), carrier (24%)], Sodium chloride, DL-methionine (99%), Calcium formate, L-threonine, ButiPearl^TM^ brand encapsulated butyric acid [composition: butyric acid as a calcium salt (50%), hydrogenated plant fat, and flavoring substances], Choline chloride (75%), Flavor, Tryptophan, Optisweet SD (sweetener), Ronozyme^®^HiPhos (Warszawa, Poland) (GT) [phytase enzyme (IUB No. 3.1.3.26): min. activity 10,000 FYT/g], Rovabio^®^ Excel AP (Warszawa, Poland) [mixture of enzymes: mainly xylanase (No EC 3.2.1.8) and β-glucanase (No EC 3.2.1.6)]. ** Nutritional value of pre-mix (in 1 kg): Calcium, 23.10%; vit. A, 4,000,000 IU; vit. D_3_, 400,000 IU; vit. E, 32,000 mg; vit. K_3_, 800 mg; vit. B_1_, 700 mg; vit. B_2_, 1600 mg; niacin (B_3_), 10,000 mg; pantothenic acid (B_5_) 6000 mg; vit. B_6_, 1400 mg; vit. B_12_, 12,000 mcg; biotin, 40,000 mcg; vit. C, 40,000 mg; folic acid, 1000 mg; iron, 30,000 mg; manganese, 14,000 mg; copper, 32,000 mg; iodine, 280 mg; cobalt, 140 mg; selenium, 50 mg; molybdenum, 200 mg; antioxidants, 3000 mg, “-” -not used.

**Table 2 animals-11-02030-t002:** Composition * and nutritive value of starter mixtures for piglets.

Item	Group I (ZnSO_4_)	Group II (ZnO)		Group III (nZnO)
**Ingredients, %**				
ZnSO_4_	0.04	-		-
ZnO	-	0.03		-
nZnO	-	-		0.03
**Nutrition Value (% in 1 kg of Feed Mixture** **)**				
Metabolizable Energy, 12.29 MJ; Crude protein, 16.51; Calcium, 0.65; P_total_, 0.50; Lysine, 1.04; Methionine, 0.35; Threonine, 0.68; Tryptophan, 0.22				
**Addition of Zn (mg in 1 kg of Feed Mixture)**				
Zinc (according to assumptions)	150	150	150	
Zinc (according to chemical analyses)	130	145	169	

* Basic composition of all mixtures: Barley, Wheat, Maize, Soya bean meal (46% CP), Oat, Rye, Rape seed meal, H-35 Fish pulp (DM 88%), Limestone, Soy oil, Monocalcium phosphate, Premix (0.5%) **, L–lysine (98%), Sodium chloride, Bilo-Cid Bufor [acidifying mixture, composition: phosphoric acid (41%), formic acid (24%), cytric acid (11%), carrier (24%)] Kemira Pro GIT SC1, medium chain fatty acids, DL-methionine (99%), L-threonine, Choline chloride (75%), Flavor, Tryptophan, Ronozyme^®^HiPhos (GT) [phytase enzyme (IUB No. 3.1.3.26): min. activity, 10,000 FYT/g] Rovabio^®^ Excel AP [mixture of enzymes: mainly xylanase (No EC 3.2.1.8) and β-glucanase (No EC 3.2.1.6)]. ** Nutritional value of pre-mix (in 1 kg): Calcium, 23.10%; vit. A, 4,000,000 IU; vit. D_3_, 400,000 IU; vit. E, 32,000 mg; vit. K_3_, 800 mg; vit. B_1_, 700 mg; vit. B_2_, 1600 mg; niacin (B_3_), 10,000 mg; pantothenic acid (B_5_), 6000 mg; vit. B_6_, 1400 mg; vit. B_12_, 12,000 mcg; biotin, 40,000 mcg; vit. C, 40,000 mg; folic acid, 1000 mg; iron, 30,000 mg; manganese, 14,000 mg; copper, 32,000 mg; iodine, 280 mg; cobalt, 140 mg; selenium, 50 mg; molybdenum, 200 mg; antioxidants, 3000 mg, “-” -not used.

**Table 3 animals-11-02030-t003:** Production results and diarrhea occurrence.

Specification	Group I (ZnSO_4_)	Group II (ZnO)	Group III (nZnO)	*p*-Value
Number of animals, heads				
28th day	100	100	100	
47th day	96	99	99	
74th day	86	96	96	
Body weight, kg				
28th day	8.2 ± 0.24	8.4 ± 0.50	8.3 ± 0.46	0.692
47th day	12.7 ^a^ ± 0.62	13.4 ^b^ ± 0.58	13.2 ^b^ ± 0.72	0.041
74th day	27.0 ^a^ ± 1.64	28.2 ^b^ ± 1.42	28.9 ^b^ ± 1.16	0.019
Daily gain, g				
28th–47th day	231 ^a^ ± 22.27	262 ^b^ ± 13.14	252 ^b^ ± 28.20	0.021
48th–74th day	528 ^a^ ± 49.30	542 ^b^ ± 41.11	586 ^b^ ± 41.45	0.023
28th–74th day	394 ^a^ ± 36.35	415 ^b^ ± 32.65	435 ^b^ ± 24.91	0.018
Daily feed intake, g				
28th–47th day	406 ± 0.05	426 ± 0.04	401 ± 0.03	0.303
48th–74th day	979 ± 0.08	975 ± 0.06	959 ± 0.07	0.807
Feed conversion rate, kg∙kg^−1^				
28th–47th day	1.71 ^A^ ± 0.12	1.66 ^A^ ± 0.09	1.54 ^B^ ± 0.08	0.004
48th–74th day	1.77 ^a^ ± 0.14	1.70 ^a^ ± 0.12	1.60 ^b^ ± 0.10	0.017
Diarrhea rate, %				
28th–47th day	6.2	1.27	1.59	
48th–74th day	2.01	0.24	0.24	
28th–74th day	4.1	0.68	0.82	

Significant differences marked within a row in small letters indicate *p* ≤ 0.05, while those marked with capital letters indicate *p* ≤ 0.01.

**Table 4 animals-11-02030-t004:** Laboratory results and zinc status.

Specification	Group I (ZnSO_4_)	Group II (ZnO)	Group III (n-ZnO)	*p*-Value
**Biochemical Indices Determined in Blood Serum**				
Total protein, g∙L^−1^	46.10 ± 4.85	46.47 ± 4.67	47.76 ± 5.92	0.724
Albumins, g∙L^−1^	24.20 ± 4.18	24.95 ± 2.21	25.66 ± 14.71	0.687
Globulins (α, β, δ), g∙L^−1^	21.90 ± 3.63	21.52 ± 3.95	22.10 ± 5.47	0.959
Glucose, mmol∙L^−1^	4.60 ± 1.33	5.73 ± 1.05	5.10 ± 1.15	0.125
Alkaline phosphatase, U∙L^−1^	250.90 ± 43.26	287.85 ± 56.29	307.52 ± 49.40	0.052
Urea, mmol∙L^−1^	4.45 ^A^ ± 0.26	4.75 ^A^ ± 0.41	2.52 ^B^ ± 0.46	0.000
Zinc, µmol∙L^−1^	14.60 ^A^ ± 2.37	28.70 ^B^ ± 5.26	17.33 ^A^ ± 3.44	0.000
Haptoglobin, g∙L^−1^	1.16 ± 0.53	1.37 ± 0.54	0.93 ± 0.43	0.168
**Zinc Balance Between 42nd to 47th Day of Life**				
Zn intake, mg kg^−1^ DM of feed	162.7 ^A^ ± 34.5	2299.2 ^B^ ± 600.7	170.0 ^A^ ± 33.3	0.000
Zn excretion, mg kg^−1^ DM of feces	125.7 ^A^ ± 18.9	1208.1 ^B^ ± 240.9	110.4 ^A^ ± 24.1	0.000
Zn retention, mg	37.0 ^A^ ± 30.7	1091.1 ^B^ ± 589.8	60.0 ^A^ ± 33.3	0.000
Retention/Intake, %	20.7 ^a^ ± 13.5	44.6 ^b^ ± 17.1	33.8 ^ab^ ± 15.8	0.025
Excretion/Intake, %	79.3 ^B^ ± 13.5	55.3 ^A^ ± 17.1	66.2 ^AB^ ± 15.8	0.007

Significant differences marked within a row with different small letters indicate *p* ≤ 0.05, while capital letters indicate *p* ≤ 0.01.

## Data Availability

The data presented in this study are available on request from the corresponding author.

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
