# Peer review of "Effect of Zinc Source and Level on Growth Performance and Zinc Status of Weaned Piglets"

_animals, 2021, doi:10.3390/ani11072030_

Round 1

Reviewer 1 Report

The revised manuscript is improved in some respects, but the following problems remain. Too few indicators were detected to explain the purpose of the study. The English language must be improved throughout the manuscript. In addition, there were formatting problems in the manuscript. For these reasons, I do not think this manuscript is suitable for publication in Animals. There are some concerns that are highlighted below.

  1. Line 11, use the full name instead of the short name (EU).
  2. The result description should be followed by a p
  3. The tables are not standardized and concise.
  4. The English language must be improved throughout the manuscript.
  5. Discussion needs to be improved.

Author Response

Response to Reviewer 1

Thank you for your comments.

  1. Line 11, use the full name instead of the short name (EU). - done
  2. The result description should be followed by a p
  3. The tables are not standardized and concise. - Tables are corrected according to requirements - we changed font size. After the first revision we changed table organization, they are shortened and now we hope that it is enough, we don't want to alter them more
  4. The English language must be improved throughout the manuscript. - English has been checked by a professional.
  5. Discussion needs to be improved. To say the true we don't know what to change, we improved language a little.

Reviewer 2 Report

Dear Authors

The manuscript entitled “Effect of Zinc Source and Level on Growth Performance, Biochemical Indices and Zinc Status of Weaned Piglets” is within the scope of the Animals journal and the results can be helpful to improve the performance. However, some flaws compromise the quality of the manuscript. There is no information in the manuscript how the Nano-Zn was produced. There are several methods for nano-mineral production. The purity of Nano-Zn, particle size, and how it was premixed or added to the diet. In table 2, 0.04%, 0.03%, and 0.03% of ZnSO4, ZnO, and nZnO were supplemented. Considering the purity of 40%, 75%, and unknown% for ZnSO4, ZnO, and nZnO, its confusing how the dietary doses could be set as 100, 2500, and 150 mg/kg? The title has to be revised, or clarify what does “biochemical incidence” means? The introduction is written well and properly discussed the issue and the aim of study. However, there are too many English errors. The discussion part is written poorly technically and grammatically.

English errors in Introduction.

Line 35 “into the use of various”

Lines 36-38 “They can modulate … efficiency and performance.” Please add references.

Line 38 Please put a comma after efficiency

Line 40 Please put a comma after plants

Line 38 please define what does “very rich” mean

Line 39 Please revise “we can find there among others …”Line 41 “animals' organisms”

Line 42 Please put a comma after development

Line 43 “gastrointestinal function through”

Line 43 Please put a comma after enzymes

Line 45-47 Id recommend: In practice, to reduce intensive diarrhea in piglets, the pharmacological doses of ZnO (1000 to 4000 mg · kg-1) are used with prescription.

Line 47 Please put a comma after however

Line 47-48 please revise. Three times of weaned pigs requirements?

Line 48 “The supplemental Zn requirement of weaned piglets”

Line 50-52 As far as I know, its not a common procedure to use high doses of ZnSO4 For pharmacological purposes. In reference [14], they only used 100 mg/kg to maintain the requirement. Please revise this sentence.

Line 54-57 not written well. This is a running sentence. please shorten the size and revise it. “on the other side,”

Line 58 please define “EU” and please remove capital D from “directive”

Line 60 “Union, zinc oxide can”

Line 61 Please put a comma after piglets

Line 61 “from June 2022”

Line 63 do you mean “apart from environmental concerns,”? however, its still unclear. please revise 63-66“

Line 64 “research centers”

Line 65 comma after availability

….

Author Response

Thank you very much for your constructive comments.

The manuscript entitled “Effect of Zinc Source and Level on Growth Performance, Biochemical Indices and Zinc Status of Weaned Piglets” is within the scope of the Animals journal and the results can be helpful to improve the performance. However, some flaws compromise the quality of the manuscript. There is no information in the manuscript how the Nano-Zn was produced. There are several methods for nano-mineral production. The purity of Nano-Zn, particle size, and how it was premixed or added to the diet. In table 2, 0.04%, 0.03%, and 0.03% of ZnSO4, ZnO, and nZnO were supplemented. Considering the purity of 40%, 75%, and unknown% for ZnSO4, ZnO, and nZnO, its confusing how the dietary doses could be set as 100, 2500, and 150 mg/kg? The title has to be revised, or clarify what does “biochemical incidence” means? The introduction is written well and properly discussed the issue and the aim of study. However, there are too many English errors. The discussion part is written poorly technically and grammatically.

In presented study we used Nano-Zn preparation produced in China as a DASHINOU, according to producer the purity of this product was 99% and it was obtained using mechanical method.

All additives: ZnSO4, ZnO, and nZnO were included to the feed mixture in % described in Tab.2 as per feed manufacturer’s description. By the way, in manufacturer’s description we found that the purity of ZnSO4 is 35% not 40% as declared originally so we introduced changes. The addition of Zn in different forms was introduced to our diets according to obtained the chemical analyzes concentrations in those additives. The level of experimental additives was adjusted to obtain acceptable - 150 mg/kg or pharmacological - 3000 mg/kg of Zn levels.

We decided to remove "biochemical incidence" from the title, it is enough to present it in Methods and Results.

English has been checked by a professional.

English errors in Introduction.

Line 35 “into the use of various” - done

Lines 36-38 “They can modulate … efficiency and performance.” Please add references. - after the next sentence we cited adequate references; 3-7

Line 38 Please put a comma after efficiency - done

Line 40 Please put a comma after plants - done

Line 38 please define what does “very rich” mean - thank you for this comment, we changed it into: " consists of a large number of products"

Line 39 Please revise “we can find there among others … - we deleted this term

Line 41 “animals' organisms” - done

Line 42 Please put a comma after development - done

Line 43 “gastrointestinal function through” - we changed it into "digestive tract"

Line 43 Please put a comma after enzymes - done

Line 45-47 Id recommend: In practice, to reduce intensive diarrhea in piglets, the pharmacological doses of ZnO (1000 to 4000 mg · kg-1) are used with prescription. - thank you for your comment, we changed this sentence

Line 47 Please put a comma after however - done

Line 47-48 please revise. Three times of weaned pigs requirements? - done

Line 48 “The supplemental Zn requirement of weaned piglets” - done

Line 50-52 As far as I know, its not a common procedure to use high doses of ZnSO4 For pharmacological purposes. In reference [14], they only used 100 mg/kg to maintain the requirement. Please revise this sentence. - that's true in cited article authors used in their experiment lower doses of Zn but in Introduction we can find the sentence: "The Zn requirement for nursery pigs given in the Nutrient Requirements for Swine (NRC, 1998) is set at 100 ppm Zn; however, the addition of 2,000 to 3,000 ppm Zn as ZnO is a common recommendation of the swine feed industry." - so why we citied it in our text, however, you are right, our citation was wrong and we removed  "or zinc sulphate (ZnSO4)" - which was wrong

Line 54-57 not written well. This is a running sentence. please shorten the size and revise it. “on the other side,”

thank you for this comment, we changed it into: Additionally, the excess of Zn is excreted in the feces which affects the environment. Moreover the excess of Zn could be accumulated in tissue (food of animal origin) which is connected with increased cellular stress and lower absorption of other micro- and macroelements.

Line 58 please define “EU” and please remove capital D from “directive” - done

Line 60 “Union, zinc oxide can” - done

Line 61 Please put a comma after piglets - done

Line 61 “from June 2022” - done

Line 63 do you mean “apart from environmental concerns,”? however, its still unclear. please revise 63-66“ - Thank you, we changed this sentence into:

Many researchers focused their attention on new sources and forms of Zn to protect environment, animals' health and quality of final products. The intention is to improve Zn utilization, availability, and absorption from the intestinal content to reduce amount of this additive in a diet.

Line 64 “research centers” - we changed this sentence

Line 65 comma after availability we changed this sentence

Reviewer 3 Report

Authors have improved the text since previous submission. However, some information is still missing, mainly on material and methods section. Also, some grammar errors need correction, as follows:

Simple summary

Line 12: Please correct the phrase "...from June 2022 so evaluating of different lower doses..." to "...from June, 2022; so, the evaluation of different lower doses..."

Abstract

Line 25: There is no need to present the abbreviation ADG since it was not repeated along abstract. Please, delete it.

Line 28: Please, correct the phrase: "prevent the accumulation of excessive this element...." to "....prevent the excessive accumulation of this element...."

Line 39: Please, correct "synbiotics" to 'symbiotics"

Introduction

Line 54: Please, standardize the use of the word "faeces/feces" along text. Authors used US Eng., so,  they should use "feces". Also, check it in lines 136, 145, 146, 160, 166, 190 and 210.

Line 68: Please, delete the extra parenthesis.

Line 75: Authors should standardize the use of word "diarrhea/diarrhoea" along text. Authors used US Eng., so they must use "diarrhea". Also, check it in line 136.

Material and Methods

Line 83: Where is located the farm? Provide this information in text.

Line 84: The diarrhea problem was solved after the findings of the proposed study. Thus, "...which was resolved by feeding strategy." should be removed.

Line 85: Please, provide the average initial BW ± sd (kg) of weaned piglets in text. 

Line 85: How many males and females? Please, provide this information in text.

Lines 87/88: Please, provide the specification about the experimental design (completely random or randomized block?) in text.

Line 89: The authors should provide at least the average humidity (± sd) and temperature levels (minimum ± sd and maximum ± sd) throughout each rearing phase.

Line 100: Authors should provide the purity of each Zn source, after the first mention of it, not only in Tables 1 and 2. For example: line 100 ("...addition of zinc sulfate (40%) ...."), line 101 ("...containing zinc oxide (75%)...") and so on.

Line 145: The fecal sampling was made how many times per day? Specify what type of indigestible marker (and its concentration) was used during  fecal collection. 

Line 150: Please, provide the information regarding the blood processing (parameters of centrifugation: speed, temperature, time; pre-treatment of samples in case of hemolysis) in text. Besides that, authors should specify in text the blood collection system (needles connect to a vacuum container or to a syringe?) and the types of tubes utilized for each variable quantified (non-additives tubes? or with which additive tubes?)

Line 151: What was the criteria of piglet selection from each experimental unit for blood collection? Please, provide it in text.

Line 154: Please, alter "In serum were determine..." to "The following analytes were determined in serum..."

Lines 160 to 162: The Zn-related quantification data are the central axis of this article. Thus, more information regarding the quantification of it should be presented in section Laboratory analysis: a) in the feed/feces (sample particle size, mineralization step program in muffle furnace, reagents used on solubilization of ash), b) in the feces (lyophilization program), c) serum (gradual drying - time and temperature).

Regarding the AAS analysis, specify the main setting parameters in text: operation module (FAAS? or GFAAS?; determinations made in duplicate or triplicate?), operating parameters (wavelength, slit width, gas type, gas flow, measuring mode - integrated Abs?, background correction?).

Statistical analysis

Line 175: Please correct the p-values with comma to dot. The same in Table 4, last row, to the right (p = 0.007). Also, correct "...significance," to "...significance." (comma to final dot)

Results

Line 202: Please, alter "...wasn´t..." to "...was not..."

Line 212: Please, correct "...differences of 23,9%..." to "...differences to 23.9%..."

lines 220/221: Please, correct the sentence "From the other side animals obtained the highest dose of Zn, more than 50% of this element excrete into environment." to "From the other side, in animals which presented the highest dose of Zn, more than 50% of this element was excreted into environment."

Discussion

Line 237: Please correct ."...reported Pei et al. [34]." to "...were reported by Pei et al. [34]."

Line 247: Please, remove the extra final dot.

Line 265: Please correct the part of sentence "...what was confirmed in own study." to "...what was confirmed in our study." 

Conclusions

Line 279: Please, correct the phrase: "prevent the accumulation of excessive this element...." to "....prevent the excessive accumulation of this element...."

Funding

Please provide the grant number of funding process.

Institutional Review Board Statement

This section is missing on the article. According to research ethics of this journal, authors must provide details of animal use approval by a research ethics committee. As a minimum, the project identification code and name of the ethics committee should be stated in this section.

Author Response

Thank you for your constructive comments.

Simple summary

Line 12: Please correct the phrase "...from June 2022 so evaluating of different lower doses..." to "...from June, 2022; so, the evaluation of different lower doses..." - done

Abstract

Line 25: There is no need to present the abbreviation ADG since it was not repeated along abstract. Please, delete it. - done

Line 28: Please, correct the phrase: "prevent the accumulation of excessive this element...." to "....prevent the excessive accumulation of this element...." - done

Line 39: Please, correct "synbiotics" to 'symbiotics" - done

Introduction

Line 54: Please, standardize the use of the word "faeces/feces" along text. Authors used US Eng., so,  they should use "feces". Also, check it in lines 136, 145, 146, 160, 166, 190 and 210. - done

Line 68: Please, delete the extra parenthesis. - done

Line 75: Authors should standardize the use of word "diarrhea/diarrhoea" along text. Authors used US Eng., so they must use "diarrhea". Also, check it in line 136. - done

Material and Methods

Line 83: Where is located the farm? Provide this information in text. - we changed the description: “The experiment was performed in field conditions at farm working in a close cycle and located in the western part of Poland, in a region with a high density of pig production. A post weaning diarrhea (PWD) was a persistent problem there.”

Line 84: The diarrhea problem was solved after the findings of the proposed study. Thus, "...which was resolved by feeding strategy." should be removed. - done

Line 85: Please, provide the average initial BW ± sd (kg) of weaned piglets in text.  - done

Line 85: How many males and females? Please, provide this information in text. unfortunately, we do not know the exact gender distribution so we can't introduce this information

Lines 87/88: Please, provide the specification about the experimental design (completely random or randomized block?) in text. - animals were divided into experimental groups according to initial body weight

Line 89: The authors should provide at least the average humidity (± sd) and temperature levels (minimum ± sd and maximum ± sd) throughout each rearing phase. - introduced, these parameters were automatically controlled at the farm.

Line 100: Authors should provide the purity of each Zn source, after the first mention of it, not only in Tables 1 and 2. For example: line 100 ("...addition of zinc sulfate (40%) ...."), line 101 ("...containing zinc oxide (75%)...") and so on. - done

Line 145: The fecal sampling was made how many times per day? Specify what type of indigestible marker (and its concentration) was used during fecal collection. - we did not use an indigestible marker. We knew exactly what amount of feed was consumed per pen and we knew exactly the daily amount of excrements so we could calculate the Zn retention as a difference between Zn intake and Zn excretion

Line 150: Please, provide the information regarding the blood processing (parameters of centrifugation: speed, temperature, time; pre-treatment of samples in case of hemolysis) in text. Besides that, authors should specify in text the blood collection system (needles connect to a vacuum container or to a syringe?) and the types of tubes utilized for each variable quantified (non-additives tubes? or with which additive tubes?)

description was completed, but we want to underline that we use only part of collected by veterinarian blood to routine health examination performed those time:

“Additionally, during routine health control performed at the farm by the veterinarian on the 47th day of life, the blood was drawn from the vena jugularis externa using 5 ml polypropylene disposable syringes (PolFast, Poland) with 16 - 19 G × 1 1/2ʼʼ needle (MIFAM). Next a portion of blood sample collected to obtain serum was transferred to tubes with pearls and was left for two hours and then centrifuged (MPW-223e laboratory centrifuge, Poland) at 2000 x g for 10 min. at room temperature. Next, for experimental needs, a part of sampling serum (from 7 randomly chosen piglets/experimental group) was taken to determine chosen parameters.”

Line 151: What was the criteria of piglet selection from each experimental unit for blood collection? Please, provide it in text. - done

Line 154: Please, alter "In serum were determine..." to "The following analytes were determinein serum..."- done

Lines 160 to 162: The Zn-related quantification data are the central axis of this article. Thus, more information regarding the quantification of it should be presented in section Laboratory analysis: a) in the feed/feces (sample particle size, mineralization step program in muffle furnace, reagents used on solubilization of ash), b) in the feces (lyophilization program), c) serum (gradual drying - time and temperature).

Regarding the AAS analysis, specify the main setting parameters in text: operation module (FAAS? or GFAAS?; determinations made in duplicate or triplicate?), operating parameters (wavelength, slit width, gas type, gas flow, measuring mode - integrated Abs?, background correction?).

this part of Methods was completed

Statistical analysis

Line 175: Please correct the p-values with comma to dot. The same in Table 4, last row, to the right (p = 0.007). Also, correct "...significance," to "...significance." (comma to final dot) - done

Results

Line 202: Please, alter "...wasn´t..." to "...was not..." - done

Line 212: Please, correct "...differences of 23,9%..." to "...differences to 23.9%..." - done

lines 220/221: Please, correct the sentence "From the other side animals obtained the highest dose of Zn, more than 50% of this element excrete into environment." to "From the other side, in animals which presented the highest dose of Zn, more than 50% of this element was excreted into environment." - done

Discussion

Line 237: Please correct ."...reported Pei et al. [34]." to "...were reported by Pei et al. [34]." - done

Line 247: Please, remove the extra final dot. - done

Line 265: Please correct the part of sentence "...what was confirmed in own study." to "...what was confirmed in our study." done

Conclusions

Line 279: Please, correct the phrase: "prevent the accumulation of excessive this element...." to "....prevent the excessive accumulation of this element...." - done

Funding

Please provide the grant number of funding process. - done

Institutional Review Board Statement

This section is missing on the article. According to research ethics of this journal, authors must provide details of animal use approval by a research ethics committee. As a minimum, the project identification code and name of the ethics committee should be stated in this section.- done

Round 2

Reviewer 1 Report

No.

Author Response

Thank you very much for your acceptance

Reviewer 2 Report

Dear Authors

There is still no information in the manuscript on how the Nano-Zn was prepared. It has been mentioned that the size of nano-ZnO is lower than 100 mm, however, no detailed information about the size range. There are several methods for nano-mineral production and it seems a pure nano-ZnO (99%) ZnO has been used in the current study (no polymers?). Nano-ZnOs size range, polymer dispersion index, zeta potential, and the specific surface area are the basic information that can be provided. The English errors decreased and the manuscript reads much better, however, there are still many English errors.

Lines 36-38 “They can modulate … efficiency and performance.” Please add references. The references of next sentence cannot cover it.

Line 38 “ This group of additives consists of a large”

Line 39 Please revise “we can find there among others …”Line 41 “animals' organisms”

Line 41 Please put a comma after element

Line 45 Please put a comma after piglets

Line 46-49 please revise. Three times of weaned pigs requirements? it means the requirement of Zn is 50 mg/kg for weaned pigs?  In the next sentence its written 100 mg/kg for weaned pigs.

Line 53 “the excess excreted Zn adversely affects the environment”

Line 56 “Therefore, the European”

Author Response

Thank you very much for your insightful comments and very helpful suggestions. We hope that our answers are enough satisfied for you.

There is still no information in the manuscript on how the Nano-Zn was prepared. It has been mentioned that the size of nano-ZnO is lower than 100 mm, however, no detailed information about the size range. There are several methods for nano-mineral production and it seems a pure nano-ZnO (99%) ZnO has been used in the current study (no polymers?). Nano-ZnOs size range, polymer dispersion index, zeta potential, and the specific surface area are the basic information that can be provided. The English errors decreased and the manuscript reads much better, however, there are still many English errors.

We forgot to introduce information about nano-ZnO product into manuscript, however we responded you partially last time. Now we introduced correction in our Material and Methods information:

"We used Nano-Zn preparation DASHINOU NANOTECHNOLOGY(CHANGZHOU) CO.,LTD., China. According to the producer the purity of this product was 99%, a mean size of nanoparticles varied between 10 and 30 nm, and it was obtained using mechanical method"

Lines 36-38 “They can modulate … efficiency and performance.” Please add references. The references of next sentence cannot cover it.

For us the next sentence just close this section and in the end we present some additives which were described in citied articles. But we decided to change our citation and moved item 3 after suggested by you sentence, where exactly we wrote about it.

Line 38 “ This group of additives consists of a large” - done

Line 39 Please revise “we can find there among others …”Line 41 “animals' organisms” - we changed it after the second revision, so it is done

Line 41 Please put a comma after element - done

Line 45 Please put a comma after piglets - done

Line 46-49 please revise. Three times of weaned pigs requirements? it means the requirement of Zn is 50 mg/kg for weaned pigs?  In the next sentence its written 100 mg/kg for weaned pigs.

Thank you for this question. We did a small mistake when we wanted to present shortly  information presented by Brugger et al. ( 2014) :

"According to the broken-line calculations shown in Figures 1 and 2, the transition from deficient to sufficient Zn supply ranged between total dietary Zn of 47.5 and 58.2 mg/kg, which was equivalent to dietary additions of Zn from Zn sulphate of 20–30 mg/kg."

 We changed our controversial sentence into: However, the allowed maximum of 150 mg · kg-1 is still higher for weaned piglet’s requirements [13].

Line 53 “the excess excreted Zn adversely affects the environment” - done

Line 56 “Therefore, the European” - done

This manuscript is a resubmission of an earlier submission. The following is a list of the peer review reports and author responses from that submission.

Round 1

Reviewer 1 Report

Dear authors,

Thank you for a very interesting article which provides very good insights and possibilities of minerals dietary usage in piglet production, after the ban of antibiotics usage in daily consumption.

Besides minor English language grammar and style editing, it would be good to include Ethical Committee approval with the protocol number for experiment with live animals (If it's possible in this case and applicable).

All the best and stay safe

Reviewer 2 Report

The present manuscript describes relative effects of dietary supplementation of 150-ppm ZnSO4 [negative control (NC)], pharmacological (3,000-ppm) ZnO [positive control (PC)], and 150-ppm nano-ZnO (nZnO; TRT) on growth performance, incidence of diarrhea, concentrations of some blood metabolites, and Zn digestibility in post-weaning piglets. This area is important scientifically as well as industrially and also conforms well to the scope of Animals. It is my regret, however, to evaluate this paper not to be suitable for publication as a ‘full’ paper because of several critical problems, including an unclear experimental design, a limited number of measurements, inadequate statistical analysis, improper organization of the manuscript, ‘poor’ English, etc. Yet, the authors might screen only significant information from the study and rewrite the manuscript in a short-paper form.

Experiment: design, statistical analysis and interpretation

“Effect of zinc source and level on ---” in the title and “The aim of the experiment was to evaluate the effect of zinc supplementation in different forms to weaners” in Line (L) 19 is too much for the content of the present experiment, because there are only three unclear dietary treatments. It is well known that pharmacological dietary supplementation of common ZnO at 2,000 to 4,000 ppm is effective for preventing and treating diarrhea and also for increasing growth performance of post-weaning piglets. It is also known in the literature that nZnO, porous ZnO, or ZnO carried by a clay mineral or encapsulated with lipid is more effective than the unprocessed, common ZnO or other inorganic Zn supplement(s) in aforementioned effects of dietary ZnO. Now that dietary ZnO supplementation is limited to 150 ppm in some countries including the EU, use of nZnO or a processed ZnO supplement is a potential alternative to pharmacological ZnO. However, relative effects of low-dose nZnO compared with those of pharmacological ZnO have not been worked out. Given these facts and the design of the present experiment, the primary aim of the present experiment could be narrowed down to evaluation of the effect of dietary supplementation of 150-ppm nZnO (TRT) vs. PC instead of the “effect of zinc source and level.” Under this aim of the experiment, comparisons between the dietary treatments may well be limited to PC vs. NC and TRT vs. NC; alternatively, comparisons between (TRT and PC):NC and TRT:PC by the contrast also could be chosen. In this context, separation of means by Duncan’s multiple-range test chosen by the authors (L190-191) is not readily agreeable.

It is an unbalanced dietary treatment that the in-feed ZnO concentration was reduced to 150 ppm from 3,000 ppm upon switching to the starter diet from the pre-starter at 47 days of age only for the PC group, with the dietary ZnSO4 and nZnO concentrations maintained for the NC and TRT diets, respectively, throughout the experiment. To be balanced, the dietary Zn supplement and its concentration should have been sustained throughout the experiment or changed to 150-ppm ZnSO4 upon switching to the starter diet for all groups. It is also a drawback of this study that blood metabolites and Zn digestibility were measured only for the pre-starter phase. Regarding the mortality, it is enigmatic why only 3% of the PC group died during the starter phase on the diet supplemented with 150-ppm ZnO while a much greater percentage (10%) of the NC group died during the same period (Table 3). As such, why the ZnO concentration for the PC starter diet was reduced, as well as what was the consequence of the change of dietary ZnO supplementation, needs to be explained convincingly. Otherwise, the starter phase may as well be excluded from the present study.

Organization of the manuscript

The present manuscript looks like a paper that has been incompletely transformed into a scientific journal article format from a thesis or dissertation. All sections, including Introduction, Materials and Methods, Discussion, Conclusions, and References, are too lengthy and unfocused compared to a limited number of variables. This paper therefore needs be shortened concisely, which is why transforming this paper to a short-paper form was suggested at the opening comment.

English

The present manuscript does meet the standard for publication as a scientific paper in terms of the ‘quality’ of English. There are numerous inaccurate, inappropriate, or awkward sentences/phrases/words including grammar errors throughout the text. The authors are advised to have the manuscript professionally language-edited if they decide to rewrite or revise the manuscript.

Others

In Materials and Methods, it looks odd that encapsulated nZnO was used in this experiment (L114-115). The authors might check if they really used encapsulated nZnO. It appears that dead animals were excluded in calculation of average daily feed intake (ADFI) and average daily gain (ADG), if mortality data are considered (Table 3). If so, it needs to be explained how ADFI and ADG were calculated. The method for measurement of Zn digestibility looks too rough; some statements regarding determination of Zn, including L176-181, are more of results than methods. Finally, blood measurements are only partially acceptable, because the blood sample was collected only from seven animals per treatment consisting of 10 experimental units or 10 pens.

In Table 3, it looks odd that the p-value for the daily gain was very low compared with the relatively high SEM:mean ratio for each group. The authors might check if the pen was the experimental unit in all variables including ADG. The feed intake, which was greater than 7 kg in every phase (period) × treatment combination, should be indicated as the daily feed intake per animal. The diarrhea rate and feces consistency (preferably fecal consistency score) need to be analyzed statistically. Overall, the authors might consult an expert in statistics for proper statistical analysis of data. Moreover, Tables 4 and 5 contain only a limited number of variables. The authors might select only significant variables from these two tables and make only one table or merge them into Table 3.

In Discussion, the authors mistook (lipid-)coated ZnO for nZnO in the statement for the studies of Wang et al. (Ref. 41; L275-277) and Kim et al. (Ref. 49; L283-286). The statement “Slightly higher concentration of TP and albumin in the group III --- health status.” in L291-293 is not supported by the result of statistical analysis. If the authors ignore the results of statistical analysis, what’s the use of such analysis?

Reviewer 3 Report

Comments and Suggestions for Authors

Comments, review of manuscript id: Animals-1206031

Title:“Effect of Zinc Source and Level on Growth Performance and Health Status of Weaned Piglets”

The manuscript describes the effect on addition of ZnO, nZnO and ZnSO4 on diarrhoea control, growth rate and health status. The results of the current study demonstrated that supplemental a commonly accepted level of nZnO (150 mg.kg-1) can promote growth and alleviate diarrhea as well as, or even better than a pharmacological dosage of conventional ZnO (3000 mg.kg-1). Although this manuscript is valuable to practical production, they are rather limited, it would be ever better if some extra data were added. The English language must be improved throughout the manuscript. In addition, there were formatting problems in the manuscript. For these reasons, I do not think this manuscript is suitable for publication in Animals. There are some concerns that are highlighted below.

  1. Title: The term “health status” used in the title is too broad. In fact, only growth performance, biochemical Indices, and Zinc status were evaluated in the manuscript, which is not indicative of health status.
  2. The simple summary and abstract sections are nearly duplicated, so it is recommended that the simple summary section should be rewritten.
  3. In general, p values for significance should be italicized.
  4. The tables in the manuscript needed to be reworked to make them more aesthetically pleasing and to remove unnecessary lines (Table 1 and Table 2). Similarly, table headers need to be centered in a uniform format (Table 4).
  5. The English language throughout the manuscript needs to be improved, especially in the abstract and results sections.
  6. For experimental diets, Group I received a standard prestarter and then a starter diet with the addition of zinc sulfate (at the level of 150 mg kg -1 in each feed type). Group II was given a prestarter feed mixture containing zinc oxide at 3000 mg kg -1 and then a starter feed with ZnO at 150 mg kg -1. Group III was fed mixtures supplemented with nZnO (at the level of 150 mg kg -1 in a prestarter and a starter feed mixture). Why did Group II add two different levels of ZnO while Group I and Group III add the same levels of ZnSO4 and nZnO? This needs to be explained in the manuscript.

Minor comments:

  1. Line37: Consider using “growth promoters” instead of “growth promotors”.
  2. Line40 and 41: Consider deleting “on”.
  3. Line44: Consider using “forms” instead of “form”.
  4. Line96: Consider using “divided into 3 experimental groups” instead of “divided on 3 experimental groups”.
  5. Line129, 138: Consider using “acid (41%), formic acid (24%), cytric acid (11%)” instead of “acid (41 %), formic acid (24%), cytric acid (11 %)”.
  6. Line 198-199: “In Table 3 are presented the growth performance and feaces evaluation data in two feeding phases. In the post-weaning period, a tendency towards diversification of BW was recorded” This result is not accurate, please rewrite it.
  7. Line205: Consider replacing “between groups” with “among groups”.
  8. Line218: “body condition (BW)” The abbreviation does not seem accurate enough.
  9. Line230: Consider replacing “Groups i” with “Groups I”.
  10. Line248: “Excretion of zinc expressed in % was the lowest in experimental groups II and III, what was unlike to amount of Zn in feces which gets into the environment” The meaning of this sentence is confusing, please rewrite it.
  11. In general, improve the English language.